# Investigation of Micromachined Antenna Substrates Operating at 5 GHz for RF Energy Harvesting Applications

**DOI:** 10.3390/mi10020146

**Published:** 2019-02-22

**Authors:** Noor Hidayah Mohd Yunus, Jumril Yunas, Alipah Pawi, Zeti Akma Rhazali, Jahariah Sampe

**Affiliations:** 1Institute of Microengineering and Nanoelectronics, Universiti Kebangsaan Malaysia, 43600 Bangi, Selangor, Malaysia; noorhidayahm@unikl.edu.my (N.H.M.Y.); jahariah@ukm.edu.my (J.S.); 2Communication Technology Section, Universiti Kuala Lumpur British Malaysian Institute, Batu 8 Sungai Pusu, 53100 Gombak, Selangor, Malaysia; alipah@unikl.edu.my; 3Electronics & Communication Engineering Department, College of Engineering, Universiti Tenaga Nasional, Jalan UNITEN-IKRAM, 43900 Kajang, Selangor, Malaysia; zetiakma@uniten.edu.my

**Keywords:** micromachined antenna, glass, RF energy harvester, bandwidth, antenna gain, MEMS, silicon, RT/Duroid 5880, dielectric permittivity, ISM band

## Abstract

This paper investigates micromachined antenna performance operating at 5 GHz for radio frequency (RF) energy harvesting applications by comparing different substrate materials and fabrication modes. The research aims to discover appropriate antenna designs that can be integrated with the rectifier circuit and fabricated in a CMOS (Complementary Metal-Oxide Semiconductor)-compatible process approach. Therefore, the investigation involves the comparison of three different micromachined antenna substrate materials, including micromachined Si surface, micromachined Si bulk with air gaps, and micromachined glass-surface antenna, as well as conventional RT/Duroid-5880 (Rogers Corp., Chandler, AZ, USA)-based antenna as the reference. The characteristics of the antennas have been analysed using CST-MWS (CST MICROWAVE STUDIO^®^—High Frequency EM Simulation Tool). The results show that the Si-surface micromachined antenna does not meet the parameter requirement for RF antenna specification. However, by creating an air gap on the Si substrate using a micro-electromechanical system (MEMS) process, the antenna performance could be improved. On the other hand, the glass-based antenna presents a good *S*_11_ parameter, wide bandwidth, VSWR (Voltage Standing Wave Ratio) ≤ 2, omnidirectional radiation pattern and acceptable maximum gain of >5 dB. The measurement results on the fabricated glass-based antenna show good agreement with the simulation results. The study on the alternative antenna substrates and structures is especially useful for the development of integrated patch antennas for RF energy harvesting systems.

## 1. Introduction

In recent years, there has been a growing interest from both academia and industry in the deployment of energy capture of ambient energy for fully autonomous powering of microdevices, using different energy harvesting techniques. These microdevices, such as sensors, actuators and so on, are extensively utilized in various daily life applications such as in wearable devices, batteryless remote controls, structural healthcare and agriculture monitoring systems, the Internet of Things (IoT), and so on [1,2,3], in which low power consumption is highly demanded. Conventionally, energy supplies for these devices are powered by chemical batteries. The battery has a limited lifetime, limited energy capacity, and requires high maintenance, while chemical leakages and waste disposal in long-term circumstances can bring environmental issues [4,5,6]. Therefore, the energy harvesting technique plays an important role in replacing the dependency of chemical batteries.

Several ambient energy sources for energy harvesting methods, such as temperature, light, radio frequency (RF) electromagnetic field, vibration, motion, electric and magnetic field, and so on, have been studied in the literature [7,8,9]. Based on the ambient energy sources, RF energy harvesting is preferred compared to other potential energy harvesting methods. RF energy is omnipresent at any time. On a daily basis, RF energy sources are consistently produced by transmission towers, Wi-Fi signals, mobile base stations, cell phones, radio broadcast stations, televisions, and so on. RF energy is also able to provide broad extended support and lifespan to sensor devices.

The typical RF energy harvesting structure includes three main parts: the antenna, impedance matching circuit and rectifier circuit. In an RF energy harvesting system, the radiated RF signals from a transmitter station are captured by the antenna and converted into alternating current (AC) voltage. The matching circuit composed of inductor–capacitor (LC) components needs to be 50-Ω matched to ensure for maximum power transferal to the matched rectifier. The rectifying circuit converts AC voltage into direct current (DC) voltage to the attached application load. However, the omnipresent RF energy has low power density at a few μW [10]. The general structure of an RF energy harvesting system with a low-input-power setup is shown in Figure 1.

An efficient receiving antenna of the RF energy harvester plays an important role in capturing the RF signal at the available frequency band. An antenna is the first and the most important element of the RF energy harvesting system, and it affects the amount of captured energy from the environment [4]. Numerous research works have been reported on the antenna for RF energy harvesting purposes to achieve appropriate characteristics, such as high efficiency [11,12,13], lower return loss [1,14,15] and an omnidirectional radiation pattern with high gain [11,12,16]. 

Most of the reported materials for the antennas designed for RF energy harvesting are made of conventional printed circuit board (PCB)-based materials such as Rogers, RT/Duroid and FR4 substrates [1,11,14,17], which is due to the established fabrication concepts from many years ago, low cost, easy fabrication and easily available material.

However, the mechanical and structural stability of the materials is low, owing to low mechanical strength, a porous structure and dependence on the coating properties of the substrates. Furthermore, these substrate materials are not compatible with integrated CMOS microcircuitry. They need wire connections, which result in low energy efficiency due to parasites and power losses. On the other hand, it is impossible to alter the conventional substrate to attain low dielectric permittivity, *ε_r_*, in order to provide better efficiency, wide bandwidth and a low radiation resistance parameter.

One possible approach to improve these parameters is by implementing silicon (Si) technology, which was already established many years ago. Si-based microfabrication in micro-electromechanical system (MEMS) technology has shown excellent potential for the formation of highly efficient sensors, and is contributing to the enhancement of various RF antennas and devices [18]. However, the high permittivity of Si substrates lowers antenna performance [19]. Some other applications of Si based in RF, such as Si-based MEMS reconfigurable antennas and devices, have been reported in [20,21], showing the high potential of the Si technology in RF circuit applications at different frequency bands.

Besides Si, glass is an appropriate material in MEMS due to its very good mechanical properties such as scratch resistance and surface stability, and high thermal insulation and optical transparency [22]. In fact, the high mechanical stability of glass, and its electrical properties, could achieve suitable RF device specification [23]. Some other applications of glass material for antennas can be found in the automotive industry for radio FM receivers and in high-speed terahertz communication systems [24,25].

Si and glass structures can be manipulated by micromachining processes. For antenna applications, the micromachining process involves the selective removal of glass or Si substrate to produce an air cavity (*ε_r_* = 1), which decreases the total *ε_r_* of the substrate system [16,19]. 

In this project, we investigate alternative concepts to replace the conventional antenna substrate, as well as different fabrication modes, in order to discover the optimal receiving antenna design for an RF energy harvester system operating at 5 GHz in the unlicensed industrial, scientific and medical (ISM) band, which is chosen as it is a free RF signal source that receives from various radio wireless devices. The proposed antenna is appropriate to be integrated with the rectifier circuit in the RF energy harvesting system, and indeed can be fabricated in a CMOS-compatible process method.

## 2. Materials and Methods

### 2.1. Antenna Design

Previously, it was reported by Yunus et al. 2018 that a Si-based bulk micromachined antenna could display improved parameters in terms of the −10 dB bandwidth, radiation pattern, gain, and size reduction [16]. In this paper, the investigation is carried out on a performance comparison between the Si-based bulk micromachined antenna, Si- and glass-based-surface micromachined antennas, RT/Duroid 5880-based PCB, and fabricated glass-based patch antenna as well. 

The investigated antenna substrates include borosilicate glass (*ε_r_* = 4.7, loss tangent tan*δ* = 0.0037), RT/Duroid 5880 (*ε_r_* = 2.2, tan*δ* = 0.0009) and single-crystal Si (*ε_r_* = 11.9, electrical conductivity of 0.00025 S/m). The substrates have thickness of *t_sub_* = 2000 μm, 1500 μm and 525 µm for glass, RT/Duroid 5880 and Si, respectively. The difference in thickness is based on the availability of material specifications in the market. A 1-µm-thick aluminium (Al) with electrical conductivity of 3.56 × 10^−7^ S/m is structured as the metal layer for the patch and the ground plane deposited on both sides of the Si and glass substrates. A standard electroplated copper of 17.5 µm provided by Rogers Corporation is structured on both sides of the RT/Duroid 5880.

The three-dimensional (3D) model and top view layout of the rectangular, slotted patch antenna are illustrated in Figure 2. The layout for each patch antenna is the same so as to fairly evaluate the radiation characteristics and the current distribution at the layout edge, whereas the geometric dimension is optimized to resonate at 5 GHz. The rectangular patch layout of the antennas is low-profile and indeed easily fixed on a flat substrate. Here, the length and width of the substrate are indicated by *L* and *W*, respectively, while patch length and width are indicated by *l* and *w*, respectively. *L_v_* is the vertical length, *L_h_* is the horizontal length and *W_s_* is the width of the slotted patch dimensions. Furthermore, the dimensions of the air cavity underneath the patch are indicated with the width *a*, and length *b*. The air cavity has a depth of 375 μm. 

Generally, the length *l* of the rectangular patch antenna is from 0.333 *λ*_0_ < *l* < 0.5 *λ*_0_, where *λ*_0_ is the free space wavelength. The substrate thickness *t_sub_* is usually from 0.003 *λ*_0_ < *t_sub_* < 0.005 *λ*_0_. However, the dimensions also can be analyzed from the transmission line model, TM (Transverse Magnetic Wave) mode [26]. Basically, in the TM mode model, the rectangular patch dimensions are calculated as in Equation (1) to Equation (3) [1],
(1)$$w=\frac{c}{2f\sqrt{\frac{\varepsilon_{r}+1}{2}}},$$
(2)$$\varepsilon_{eff}=\frac{1}{2}\left(\varepsilon_{r}+1\right)+\frac{1}{2}\left(\varepsilon_{r}-1\right)\left[\frac{1}{\sqrt{1+12\left(\frac{t_{sub}}{w}\right)}}\right],$$
(3)$$l=\frac{c}{2f\sqrt{\varepsilon_{eff}}}-0.824t_{sub}\left(\frac{\left(\varepsilon_{eff}+0.3\right)\left(\frac{w}{t_{sub}}+0.264\right)}{\left(\varepsilon_{eff}-0.258\right)\left(\frac{w}{t_{sub}}+0.8\right)}\right),$$
where *c* is the speed of light (3 × 10^8^ m/s), *f* is the operating frequency, and *ε_eff_* is the effective *ε_r_* index of the substrate. This model shows good physical insight, but is less accurate. Resolution by property optimization in the return loss parameter graph is performed by amending the dimension values using the parametric sweep function in Computer Simulation Technology Microwave Studio (CST-MWS).

### 2.2. Fabrication of Micromachined Antennas

Figure 3 illustrates the cross-sectional view of the micromachined antenna structure and the materials used in the experiment. Two types of micromachined antenna structure, that is, the surface micromachined structure and bulk micromachined structure, are investigated using CST-MWS analysis, while Si and glass are used as the substrates.

#### 2.2.1. Surface Micromachined Antenna

The surface micromachining is the process method to create an electromechanical structure on the surface of the substrate [23,27]. Here, the Si or glass substrate materials are fabricated using a surface micromachining method, in which the antenna pattern is fabricated on top of the substrate surface. The side views of the structures are illustrated in Figure 3a,c. 

Initially, a thin metal layer of 1 µm is sputtered on the top of the substrates. Then, a photolithography process is used to pattern the metal patch, followed by a metal etch process. Finally, a metal ground plane is created by sputtering 1-µm-thick Al on the bottom of the substrates. 

#### 2.2.2. Bulk Micromachined Antenna

Bulk micromachining is the process to create the 3D structure in the bulk substrate. This process is necessary because the substrate thickness and the dielectric permittivity *ε_r_* of the substrate influence the performance of the antennas [16,19,23]. 

As the Si substrate has a high *ε_r_* index, an appropriate substrate thickness of the Si is required to decrease the *ε_r_* index. A thick Si substrate and a high *ε_r_* index excite surface waves that cause low efficiency and attenuate the radiation pattern. 

This process involves the etching of a portion of Si or glass substrate to create an air cavity between the substrate and ground plane electrodes, as illustrated in Figure 3b. Here, a 525-µm-thick Si substrate is anisotropically etched to create a cavity with a depth of 375 µm. Wet chemical etching by potassium hydroxide (KOH) etchant liquid is used in the process, while 1-µm-thick Al is sputtered at the top side of the substrate. The geometric layout of the radiator patch antenna is then patterned by a photolithography process on the top layer of the Si substrate, followed by a metal etch process. Underneath the Si substrate, a sputtered Al layer with the support of a hard transparent plastic sheet is attached to the silicon substrate as a metal ground plane. 

## 3. Results and Discussion

### 3.1. Simulation Results

The optimized dimensions are based on the first resonant frequency at 5 GHz of the *S*_11_ parameter plot. The dimensions of these antennas are optimized using CST-MWS software. Each patch antenna is characterized to have a 50-Ω input impedance. Here, by adjusting vertical length *L_v_*, horizontal length *L_h_* and slot width *W_s_* of the slotted patch dimensions, length *L* and width *W* of the substrate dimensions, as well as length *l* and width *w* of the rectangular patch dimensions, the optimized dimensions of the antennas are obtained within the specification substrates chosen. The variation of the antenna design parameter on various substrates is due to different ε_r_ values, substrate loss and electrical conductivity [28]. Therefore, each substrate considers different dimensions to obtain the targeted resonating frequency, while the dimensions of *a*, *b* and *t_air_* are based on the air cavity structures of the Si substrate. 

The studied dimension parameters are presented in Table 1. It reveals that the glass-surface micromachined antenna has the smallest dimension size of the antennas. In fact, the miniature size of the antenna makes it more compatible to be integrated into microcircuitry than other antennas.

The obtained simulated return loss results are plotted in Figure 4. From the simulation, it is shown that the return loss *S*_11_ parameter is less than −10 dB for all apart from the Si-surface micromachined antenna, which is due to the high *ε_r_*. The *S*_11_ parameter below −10 dB indicates the range where the antenna can operate properly.

Thus, an improvement of the Si structure is achieved by reducing the thickness and creating an air cavity in a definite ratio, which result in the *S*_11_ level dropping below −10 dB. This antenna structure is achieved by a bulk micromachining process. It can be seen that the *S*_11_ parameter of the Si-bulk micromachined antenna is improved by 33.5% compared to the Si-surface micromachined antenna. 

Further investigation is conducted by replacing the Si-based-surface micromachined antenna substrate with glass. Here, the glass substrate is selected because of its low *ε_r_*. The results show that the *S*_11_ parameter of the glass-surface micromachined antenna is increased by 55.1% compared to the Si-surface micromachined antenna. The antenna comparison with the conventional PCB RT/Duroid 5880-based antenna shows that the *S*_11_ parameter of the glass-surface micromachined antenna improves by 5.6%.

*S*_11_ and voltage standing wave ratio (VSWR) are significant parameters to specify the −10 dB bandwidth parameter. The bandwidth parameter is accomplished based on the return loss of less than −10 dB and the VSWR of less than 2. From the simulation, Si-bulk micromachined, RT/Duroid 5880-based and glass-surface micromachined antennas performed with VSWRs of less than 2. Accordingly, the parameter of −10 dB bandwidth was not present, and the VSWR was larger than 2 for the Si-surface micromachined antenna, which corresponds to its high *ε_r_* index substrate in the structure.

A wide bandwidth range achieved from 100 MHz to 2.45 GHz is a required parameter in the ISM band standard. Thus, the −10 dB bandwidths of the glass-surface micromachined antenna of approximately 117 MHz (from 5.0644 to 4.9474 GHz), and of the RT/Duroid 5880-based antenna of approximately 115 MHz (from 5.063 to 4.948 GHz), are found acceptable. On the other hand, a narrow bandwidth of 32 MHz (from 4.981 to 5.013 GHz) is seen for the Si-based bulk micromachined antenna. The glass-surface micromachined and RT/Duroid 5880-based antennas are under the circumstance of the standard wide-frequency range. It is therefore concluded that the antenna designed with a substrate of low *ε*_r_ index contributes to a good bandwidth parameter performance. 

The simulated *E* and *H* planes of 2D far-field radiation patterns of the antennas are shown in Figure 5. In the *E*-plane, the antennas show a linear top symmetrical and omnidirectional beam pattern. In the *H*-plane, the Si-surface micromachined antenna shows a nonhomogeneous minor radiation pattern, and the other three antennas show a top omnidirectional beam pattern. It can be observed that there is a major radiation pattern at the top side of both *E* and *H* planes. Thus, these antennas are focused for the radiation beam from the top radiator patch.

The simulated realized gain and the directivity parameters of the RT/Duroid 5880-based antennas are higher compared to the other antennas. However, the parameters of the glass-surface micromachined antenna can be considered better with a gain parameter of more than 5 dB as well as a directivity parameter more than 3 dBi, good enough for deliberating the radiation beam in a desired position. The higher directivity of the RT/Duroid 5880-based and the micromachined Si bulk antennas compared to the glass-surface micromachined antenna illustrates that both antennas are more effective at focusing the energy captured from a static position in a precise direction.

### 3.2. Validation

To validate the simulation analysis, the glass substrate was selected to undergo the fabrication process owing to the most appropriate performance achieved in the simulation analysis among other substrates. Therefore, the borosilicate glass-based-surface micromachined antenna was fabricated using standard MEMS processes. The fabricated antenna was then measured in an anechoic chamber environment on an indoor far-field range. The measured antenna property was then compared with the simulation. The fabricated antenna from the top- and bottom-plane layouts is as shown in Figure 6. The final structural layout of the glass substrates was following the dimensions optimized at 5 GHz. The Al top feed line patch and Al bottom ground plane were connected to a coaxial SMA (SubMiniature version A) connector.

The measured return loss *S*_11_ and radiation pattern parameters of the antenna are shown in Figure 7 and Figure 8, respectively. It can be observed that the measured resonant frequency is shifted to the left by approximately 50 MHz from the targeted 5 GHz frequency. In addition, the measured −10 dB bandwidth result also presented significant changes, more than twice as wide as the simulated result. The shifting frequency and the change in bandwidth are possibly due to the inaccurate cutting edge of the substrate structure and an imprecise geometric dimension of the pattern patch during the fabrication process. Correspondingly, the slight inaccuracy and imprecise adjustment in the geometric dimensions cause changes in the capacitive and resistive loss parameters of the substrate. The measured return loss is slightly decreased, and a significant enhancement in the −10 dB bandwidth is observed. A comprehensive analysis on the comparison between simulation and measurement results of the fabricated antenna is summarized in Table 2. The VSWR of the fabricated antenna is within the range of 1 to 2, which meets the simulated results. 

The *E* and *H* planes of 2D far-field radiation patterns of the measured antennas displayed minor changes compared to the simulated ones. However, it is observed that the bottom side of the radiator patch of the measured glass-surface micromachined antenna pattern is slightly wider than the simulated ones. The illustrated pattern is possibly influenced by the wide bandwidth and high gain parameters obtained at the operating frequency.

## 4. Conclusions

An investigation on antennas with various substrates and structures was performed for improved RF energy harvesting. Initially, the geometry of the antennas was designed and optimized to be suitable for capturing RF energy at a frequency of 5 GHz. The antenna characteristics were analyzed by comparing the effects of the substrate material and the structure manipulated through the surface and bulk micromachining processes. The results showed that the Si-based-surface micromachined antenna does not have appropriate properties for antenna applications. However, by creating an air gap between the substrate and the ground plane, the *ε_r_* index can be improved, hence enlarging the bandwidth, gain and the directivity parameters, and reducing the return loss by approximately 33.5%. Further investigation using a glass-based-surface micromachined antenna revealed that the antenna designed with a substrate of low *ε_r_* index contributes to a good bandwidth parameter performance. The *S*_11_ parameter showed a curve profile similar to that of a conventional RT/Duroid-based antenna with a return loss increase of approximately 55.1% compared to the Si-based-surface micromachined antenna. The analysis of the fabricated glass-based micromachined antenna showed a good agreement with the simulation. It is concluded that the glass-based antenna can be considered as an alternative front-end device for an integrated RF energy harvesting system. 

## Figures and Tables

**Figure 1 micromachines-10-00146-f001:**
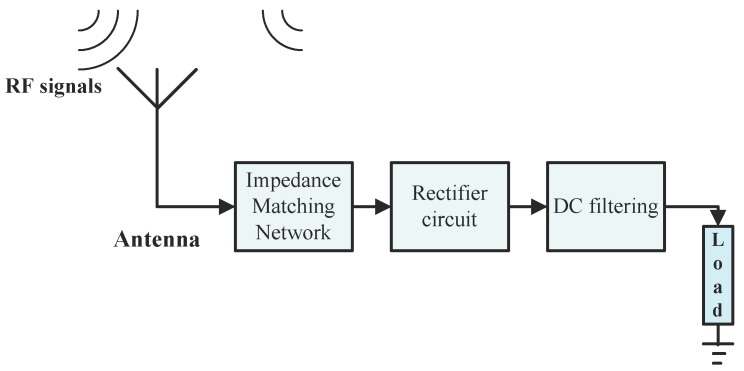
Structure diagram of radio frequency (RF) energy harvesting system.

**Figure 2 micromachines-10-00146-f002:**
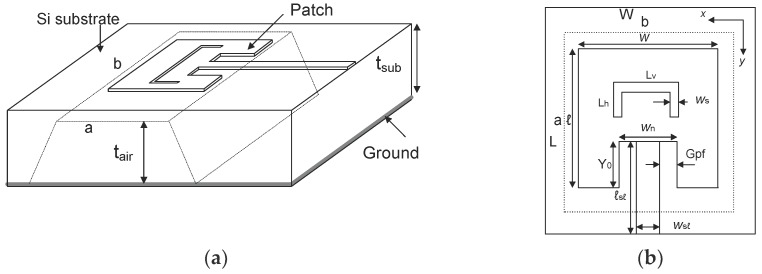
Antenna design: (**a**) the 3D model of the antenna; (**b**) the geometrical layout of the patch antenna.

**Figure 3 micromachines-10-00146-f003:**
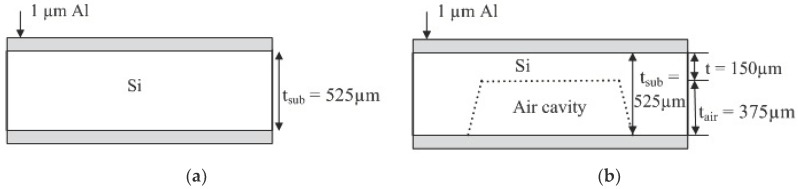
Cross-sectional view of the micromachined antenna structures: (**a**) micromachined Si-surface; (**b**) micromachined Si-bulk; (**c**) micromachined glass surface.

**Figure 4 micromachines-10-00146-f004:**
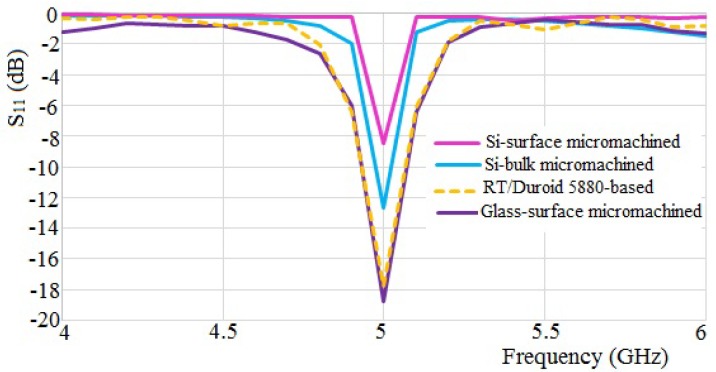
Comparison of simulated return loss.

**Figure 5 micromachines-10-00146-f005:**
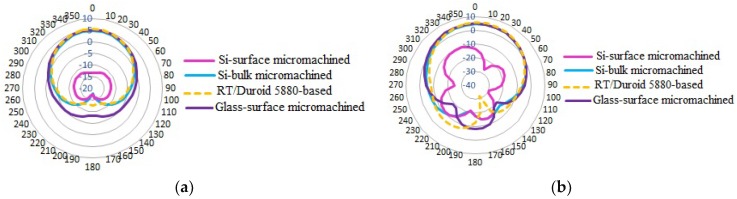
Simulated far-field radiation pattern: (**a**) *E*-field (*y*–*z* plane); (**b**) *H*-field (*x*–*z* plane).

**Figure 6 micromachines-10-00146-f006:**
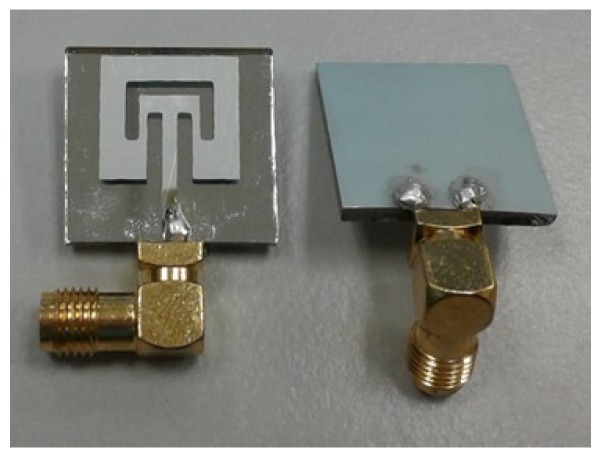
Fabricated glass-surface micromachined antenna.

**Figure 7 micromachines-10-00146-f007:**
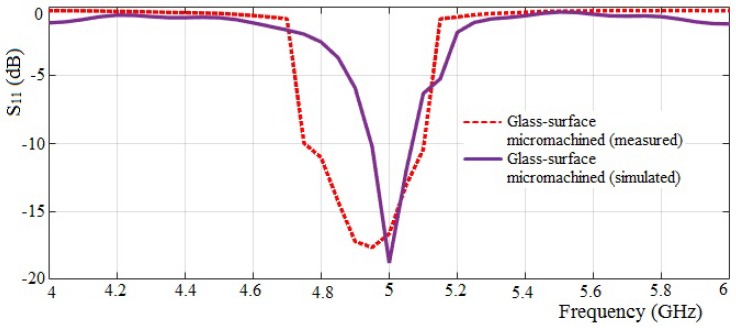
Comparison of measured and simulated return loss of glass-based-surface micromachined antennas.

**Figure 8 micromachines-10-00146-f008:**
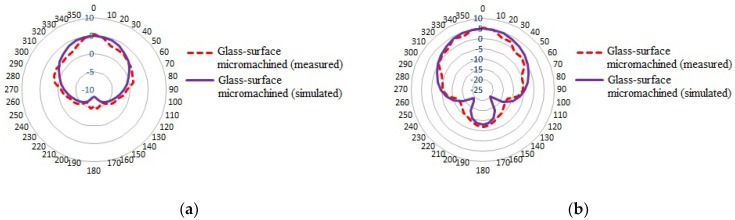
Comparison of measured and simulated far-field radiation patterns of glass-based-surface micromachined antenna: (**a**) *E*-field (*y*–*z* plane); (**b**) *H*-field (*x*–*z* plane).

**Table 1 micromachines-10-00146-t001:** Dimensions and structures of antennas optimized at 5 GHz.

Antenna/Dimension (mm)	*L*	*W*	*l*	*w*	*a*	*b*	*t_air_* (μm)
Micromachined Si surface	49	40	39.5	33	0	0	0
Micromachined Si bulk	30	27	17	17	20.46	20.46	375
RT/Duroid-based	26.99	27.5	16.98	23	0	0	0
Micromachined glass surface	19	19	10	15.5	0	0	0

**Table 2 micromachines-10-00146-t002:** Summary of antenna characteristics optimized at 5 GHz.

Antenna	*S*_11_ (dB)	VSWR	−10 dB Bandwidth (MHz)	Realized Gain (dB)	Simulated Directivity (dBi)
Si-surface micromachined **(simulated)**	−8.45	22	n/a	−7.8	3.628
Si-bulk micromachined **(simulated)**	−12.7	1.6	32	4.754	4.354
RT/Duroid 5880-based **(simulated)**	−17.75	1.3	115	5.555	7.195
Glass-surface micromachined **(simulated)**	−18.8	1.2	117	5.022	3.81
Glass-surface micromachined **(measured)**	−17.655	1.26	340	5.379	n/a

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
