# Peer review of "Investigation of Micromachined Antenna Substrates Operating at 5 GHz for RF Energy Harvesting Applications"

_micromachines, 2019, doi:10.3390/mi10020146_

Round 1

Reviewer 1 Report

The paper provides a comparison between different antenna structures for RF Energy harvesting. One thing lacking in the paper is experimental data with respect to most of the structures. While the paper only provides experimental data with respect to glass based surface micromachined antenna, it does not provide any detail about the other 3 structures. Please discuss if you have any supporting data with respect to the other structures.

Also, a short discussion is needed to tell the readers how the authors arrived at the dimensions of the antennas.

Author Response

Dear Reviewer1,

We would like to thanks for your valuable comments that give a significant impact on the quality of our paper. Some revisions have been made in our submitted revised manuscript. All the points raised in your comment have been discussed, as listed below:     

Point 1: The paper provides a comparison between different antenna structures for RF Energy harvesting. One thing lacking in the paper is experimental data with respect to most of the structures. While the paper only provides experimental data with respect to glass based surface micromachined antenna, it does not provide any detail about the other 3 structures. Please discuss if you have any supporting data with respect to the other structures.

Response 1:

1.   Thank you for the valuable comments and suggestions

2.   In this paper, our discussion is focussed on the theoretical analysis of the antenna performance on the various substrate.

3.   The experimental data is used to validate the analysis which is in respect to the glass substrate that shows the most appropriate performance among other substrates, event tough its permittivity is higher than RT/Duroid 5880.

4.   This explanation is added in the revised manuscript (please refer the Section 3.2, Line 250-253)

5.   In general, the antenna on low ɛr index substrate shows a significant performance of the wave propagation as reported in references [22], and additional reference [24]. Here, RT/Duroid 5880 has been the common substrate material for RF antenna device. The glass substrate has ɛof 4.7 while silicon has ɛr 10 which can be fabricated in CMOS compatible process [19]. Here, by introducing glass based surface micromachined antenna, the integrated circuit performance in terms of the wave propagation can be improved compared to the Si substrate based antenna.

6.   Furthermore, the performances of the fabricated devices on various substrates including its comparison are expected to be reported in the next paper.

Point 2: Also, a short discussion is needed to tell the readers how the authors arrived at the dimensions of the antennas.

Response 2:

1.   The dimension of these antennas is optimized at 5 GHz at the first resonance of the S11 parameter plot. This simulation was done using CST-MWS.

2.   Here, we obtained the dimension of the antenna by adjusting vertical length Lv, horizontal length Lh and slot width Ws of the slotted patch dimensions; length L and width W of the substrate dimensions as well as length l and width w of the rectangular patch dimensions, which is due to different ɛr, substrate loss and electrical conductivity. The dimensions of a, b and tair are based on the dimension of the air cavity of the Si substrate.

3.   All the parameters are selected in the simulation parameter to meet the targeted operating frequency at 5 GHz.   Therefore, each substrate is considered to have different antenna dimension to obtain the targeted resonating frequency.

4.   Short discussion to describe the justification of dimension choice is added in the revised manuscript. (please refer Section 3.1. Line 184-193)

Reviewer 2 Report

Review comments for manuscript entitled “Investigation of Micromachined Antenna Substrates Operating at 5 GHz for RF Energy Harvesting Application”.

The authors investigate alternative concepts to replace the conventional antenna substrate and fabrication modes in order to discover the optimal receiving antenna design for RF energy harvester system.

The attracting points is most of the references listed were up-to-date and properly discussed.

In general, this paper lack of novelty and the authors didn’t clarify the application and significance for this research.

More detailed and clear demonstration of experiment and validation must be provided. Scientific soundness must be improved.

For the table and figure, the authors only summarized the data trend without explanation of experimental data from sound theoretical view. Please provide more detailed theoretical explanation about the data in table and figures.

Author Response

Dear Reviewer2,

We would like to thanks for your valuable comments and suggestions that give a significant impact on the quality of our paper. Some revisions have been made in our submitted revised manuscript. All track changes and the amendments are marked with GREEN colour. All the points raised in your comment and suggestions have been discussed. The detailed responses are given below:

Point 1-3:

Review comments for the manuscript entitled “Investigation of Micromachined Antenna Substrates Operating at 5 GHz for RF Energy Harvesting Application”.

 The authors investigate alternative concepts to replace the conventional antenna substrate and fabrication modes in order to discover the optimal receiving antenna design for RF energy harvester system.

The attracting points is most of the references listed were up-to-date and properly discussed

Response 1-3: Thank you very much for these comments

Point 4: In general, this paper lack of novelty and the authors didn’t clarify the application and significance for this research.

Response 4:

Thank you very much for the comments.

We agree that the study of the substrate material such as Teflon, RT/Duroid as well as glass and silicon substrate have been done since many years ago.      

Here, we study the implementation of some substrate material for RF      antenna which is appropriate to be integrated with rectifier circuit in the energy harvesting system and indeed can be fabricated in CMOS      compatible process method.

So far of our knowledge, there is a few study regarding the use of      glass substrate for RF antenna application. Therefore there are still many      aspects have to be studied to understand the effect of the material for   better performance

These statements are added in the revised manuscript (please refer Section      1, Line 94-95)

Point 5: More detailed and clear demonstration of experiment and validation must be provided. Scientific soundness must be improved.

Response 5:

1.   Thank you very much for the suggestions

2.   In this study, the experimental analysis is used to validate the theoretical analysis

3.   We chose the glass substrate for experimental analysis due to the most appropriate material among other studied materials in our simulation.

4.   The detail explanation on the experimental demonstration is added in the revised manuscript (please refer 3.2, Line 250-253)

5.   We also have added Equations in Line 129 to 131 to describe the patch dimension theoretically

Point 6: For the table and figure, the authors only summarized the data trend without explanation of experimental data from sound theoretical view. Please provide more detailed theoretical explanation about the data in table and figures.

Response 6:

1.      Thank you for this comment and suggestion.

2.      Some results have been discussed and  explained from the theoretical view as found in Line 124 to 134 for Figure 2, Line 202 for S11 Level Figure 4 and Table 2, Line 216 for bandwidth range Table 2,  Line 243 for Gain parameter Table 2 and  244 for directivity parameter Table 2, and 273 for VSWR range Table 2

3.      More explanation regarding the data from other table and figure is added (please refer Section 3.1, Line 184-193 and Section 3.2 Line 264-270)

Reviewer 3 Report

The manuscript presents the design of micromachined antennas for RF energy harvesting operating at around 5 GHz with different substrates and their influence to the performances. The paper is clear and well organized.

Several things need to be addressed before acceptance:

Page 2, Line 67, the sentence is not completed, please complete the sentence.

Figure 2 (a) is not clear enough to illustrate the antenna structure. The dimension parameters used in Figure 2 (b) should be briefly explained in the context. The parameters/symbols listed in Table 1 should be consistent with those in Figure 2 (b). It is a little confusing, since they are slightly different. 

It may worth explaining why the three thicknesses for the substrates are chosen for glass, RT/Duroid 5880 and Si. 

For the bulk micromachined antenna, what is the thickness of the bottom Al adhesive solid sheet? Will it encounter any fabrication issue?

The 50 MHz shift from 5GHz target frequency of the fabricated antenna with glass substrate is explained in the Section 3.2. However, the other discrepancies need more explanation. Eg. the bandwidth at -10 dB doubles the simulation results. 

Why did the authors only fabricate the antenna with glass substrate? Won't it be more convincing when all the simulated antennas get fabricated and measured?

Author Response

Dear Reviewer3,

We would like to thanks for your valuable comments and suggestions that give a significant impact on the quality of our paper. Some revisions have been made in our submitted revised manuscript. All track changes and the amendments are marked with GREEN colour in the manuscript. All the points raised in your comment and suggestions have been discussed. The detailed responses are given below:

Point 1: Page 2, Line 67, the sentence is not completed, please complete the sentence.

Response 1:

Thank you very much for the suggestion.

We have revised the paragraph sentence with the completed sentence by adding word ‘gain’ in the last sentences. (please refer Line 67

Point 2: Figure 2 (a) is not clear enough to illustrate the antenna structure. The dimension parameters used in Figure 2 (b) should be briefly explained in the context. The parameters/symbols listed in Table 1 should be consistent with those in Figure 2 (b). It is a little confusing, since they are slightly different.

Response 2:

1.  Thank you very much for the valuable suggestion.

2.   We have revised Figure 2a and 2b for the consistency of the symbol

3.   We added the description of the dimension parameter in Line 116 to 120

 Point 3: It may worth explaining why the three thicknesses for the substrates are chosen for glass, RT/Duroid 5880 and Si.

Response 3:

1.      In this study, our objective is to analyze the performance of the patch antenna at the targeted frequency range.

2.      The choice of the substrate specification is based on the material specification available in the market and in our lab.

3.      We have added a brief explanation in the revised manuscript. (please refer Section 2.1, Line 107-108)

Point 4: For the bulk micromachined antenna, what is the thickness of the bottom Al adhesive solid sheet? Will it encounter any fabrication issue?

Response 4:

1.      The thickness of the bottom metal plays no significant influences to the antenna performance. It acts only for grounding.

2.      More important is the strength of the material to cap the cavity. In this study, we used 1 um Al layer attached on a plastic sheet to cap the air gap underneath the silicon substrate.

3.      There is no significant fabrication issue as the Al layer is deposited on plastic sheet an attached to the silicon substrate.

4.      We have added the information to response the mentioned issue in the revised manuscript. (please refer subsection 2.2.2, Line 180-181)

Point 5: The 50 MHz shift from 5GHz target frequency of the fabricated antenna with glass substrate is explained in the Section 3.2. However, the other  discrepancies need more explanation. Eg. the bandwidth at -10 dB doubles the simulation results.

Response 5:

1.      It can be observed that the measured resonant frequency is shifted to the left of approximately 50 MHz from the targeted 5 GHz frequency. Correspondingly, the measured -10 dB bandwidth result also presented significant changes of nearly triple wider than the simulated result.

2.      The shifting frequency and the -10 dB bandwidth changes are possibly due to the inaccurate cutting edge of the substrate structure during fabrication that affects to an imprecise geometric dimension of the pattern patch.

3.      Therefore, the change in the dimension will effect on the changes in the capacitive, resistive loss parameters of the substrate

4.      This statement is included in revised manuscript (please refer 3.2 Line 265-270)

Point 6: Why did the authors only fabricate the antenna with glass substrate? Won't it be more convincing when all the simulated antennas get fabricated and measured?

Response 6:

1.      In this paper, our discussion is focussed on the theoretical analysis of the antenna performance on the various substrates.

2.      The experimental data is used to validate the analysis which is in respect only to the glass substrate. This is due to the most appropriate performance obtained by glass based antenna among other substrates. Even though its permittivity is higher than RT/Duroid5880. 

3.     On the other hand, we consider also the time constraint. (please refer 3.2, Line 250 to 253)

Round 2

Reviewer 2 Report

Please add some more description on the appliation of this research.

Author Response

Point 1:

Please add some more description on the application of this research.

Response 1:

Thank you very much for the 2nd suggestion.

In this paper, we focus the study on the implementation of a various substrate including Silicon and Glass material for use as the antenna substrate for energy harvesting.

More description on the implementation of silicon and glass material for RF devices and application have been added in Line 83-86 and Line 90-92, respectively.

More description about the application of RF      antenna in unlicensed industrial, scientific and medical has been also added in Line 99.

We have also included more references for more information regarding the application of the substrate for RF application in the References [20] [21] and [24][25]. Therefore we have rearranged the new reference list, as [20] [21] [22] [23] [24] [25] [26] [27] and [28]
